

# Anti-leukemic effect of menthol, a peppermint compound, on induction of apoptosis and autophagy

Mashima Naksawat[1], Chosita Norkaew[1], Kantorn Charoensedtasin[1], Sittiruk Roytrakul[2] and Dalina Tanyong[1]

[1] Department of Clinical Microscopy, Faculty of Medical Technology, Mahidol University, Nakhon Pathom, Thailand
[2] Functional Proteomics Technology Laboratory, Functional Ingredients and Food Innovation Research Group, National Center for Genetic Engineering and Biotechnology, National Science and Technology for Development Agency, Pathum Thani, Thailand

## ABSTRACT

**Background:** Menthol, a natural compound in peppermint leaves, has several biological activities, including antioxidant, anti-inflammatory, antiviral, antibacterial and anticancer properties. This study revealed the anti-leukemic effects and its underlying mechanisms of the menthol related apoptosis signaling pathway and autophagy in both NB4 and Molt-4 leukemic cell lines.

**Methods:** Both leukemic cells were treated with menthol in various concentration. Cell viability was assessed using MTT assay, whereas apoptosis and autophagy were analyzed by flow cytometry using Annexin V-FITC/PI and anti-LC3/FITC antibodies staining, respectively. Apoptotic and autophagic related gene and protein expression were detected using RT-qPCR and western blot analysis, respectively. Moreover, STITCH database was used to predicts the interaction between menthol and proposed proteins.

**Results:** Menthol significantly decreased cell viability in NB4 and Molt-4 cell lines in dose dependent manner. In combination of menthol and daunorubicin, synergistic cytotoxic effects were observed in leukemic cells. However, there was a minimal effect found on normal, peripheral blood mononuclear cells (PBMCs). Moreover, menthol significantly induced apoptosis induction *via* upregulation of caspase-3, BAX, p53 and downregulation of MDM2 mRNA expression. Autophagy was also induced by menthol through upregulating ATG3 and downregulating mTOR mRNA expression. For protein expression, menthol significantly increased caspase-3 whereas decreased mTOR in both leukemic cells. Conclusions. These results suggest that menthol exhibits cytotoxic activities by inhibition of cell proliferation, induction of apoptosis and autophagy through activating the caspase cascade, altering BAX and p53/MDM2, and regulating autophagy *via* the ATG3/mTOR signaling pathway.

## INTRODUCTION

Leukemia is one type of hematological malignancy, which is defined as the impairment of hematopoiesis resulting in the uncontrolled and rapid proliferation of abnormal

Corresponding author
Dalina Tanyong,
dalina.itc@mahidol.ac.th

leukocytes. In 2022, there were 60,650 estimated new cases and 24,000 deaths of leukemia worldwide (*Siegel et al., 2022*). Among a lot of treatment approaches, chemotherapy is the main therapeutic for both myeloid and lymphoid leukemia, which is used to reduce the number of leukemic cells. However, there are shown adverse effects including pain, hair loss, nausea, bleeding, and weakened immune system, which may be related to the disease relapsing in patients. Therefore, the development of novel therapeutic agents with less toxicity is needed. An alternative treatment with herbal medicines is mostly focused on several research studies, since using natural products has shown the potential effects against several cancer cell lines.

Peppermint (*Mentha piperita*) has been used as traditional medicine in several conditions such as heartburn, migraines, and irritable bowel syndrome (*Almatroodi et al., 2021*). Terpenes, a class of natural products, are the most compound found in peppermint leaves which contain monoterpenes (52%), sesquiterpenes (9%), and others (*Trevisan et al., 2017*), however, menthol is the major constituent among terpenes that exhibits various biological activities, such as anti-inflammatory (*Rozza et al., 2014*), antiviral (*Taylor et al., 2020*), antibacterial (*Desam et al., 2019*), and anticancer activities in several cancer types, including bladder cancer, prostate cancer, liver cancer, skin cancer, and myeloid leukemia (*Desam et al., 2019*). Menthol increases mitochondrial membrane depolarization through TRPM8 channel leading to cell death in human bladder cancer T24 cells (*Li et al., 2009b*) and enhances anti-tumor effects by downregulating cytochrome P450 3A4 (CYP3A4) in human hepatocellular carcinoma HepG2 cells (*Nagai et al., 2019*). Besides, menthol can inhibit cell growth, cell cycle, and cell migration of prostate cancer DU145 cells *via* downregulating the focal-adhesion kinase (FAK) pathway (*Wang et al., 2012*). Menthol also induces G2/M phase arrest and inhibits hyaluronidase activity in human epidermoid carcinoma A431 cells, skin cancer, which lead to cancer cell death (*Fatima et al., 2021*). In 2020, *Kim et al. (2012)* found that menthol inhibit tumor growth of prostate cancer PC-3 cells by inducing G2/M arrest and decreasing polo-like kinase (PLK1) associated with DNA replication. Moreover, *Lu et al. (2006)* revealed that menthol also upregulates $Ca^{2+}$ in human promyelocytic leukemia HL-60 cells associated with the induction of menthol-induced cell death.

Nowadays, programmed cell deaths are important molecular mechanisms, which play an important role in cancer therapy. Several novel anticancer agents showed that apoptosis and autophagy are the main signaling pathways leading to cancer cell death (*Mishra et al., 2018*). Mostly novel agents revealed that their underlying mechanisms associate with the caspase activation which enhances apoptosis. Also, autophagy has been reported as one mechanism behind (*Deesrisak et al., 2021a*). However, there is still lack of evidence involved in menthol-induced leukemic cell death as an anti-leukemic agent and its mechanism. Therefore, the present work investigates the effects of menthol on NB4 and Molt-4 leukemic cells on cell viability, cell apoptosis, autophagy and observed the synergistic effects of menthol and chemotherapy medicines, daunorubicin. Moreover, we examined the underlying possible mechanism of menthol associated with apoptosis and autophagy.

## MATERIALS AND METHODS

### Chemical and reagents

Menthol (Sigma-Aldrich, Schnelldorf, Germany, Cat No. 63660) was purchased from Sigma Aldrich (St. Louis, CA, USA) and dissolved in DMSO for storage at 4 °C. Moreover, RNase A and propidium iodide (PI) were purchased from Sigma Aldrich (St. Louis, CA, USA). 3-(4-dimethylthiazol-2-yl)-2-5-diphenyl tetrazolium bromide (MTT) and Annexin V-FITC apoptosis assay kit was obtained from Invitrogen (Walthem, MA, USA) and BD Bioscience (Palo Alto, CA, USA), respectively. Autophagy LC3-II antibody-based assay kit was purchased from Luminex Corporation (Austin, TX, USA). All other reagents used in the experiments were commercially analytical grade and available.

### Leukemic cell culture

Human acute promyelocytic and acute T-lymphocytic leukemic cell lines, NB4 and Molt-4, were purchased from Cell Line Service GmbH (Eppelheim, Germany). Leukemic cells were cultured in RPMI-1640 medium supplemented with 10% fetal bovine serum (FBS) and 1% penicillin-streptomycin as an antibiotic (Gibco Life technologies, Waltham, MA, USA) in humidified incubator at 37 °C with 5% $CO_2$.

### Isolation of peripheral blood mononuclear cells

Blood collection was collected with the ethical approval of Mahidol University Central Institutional Review Board (MU-CIRB) (approvals no. MU-CIRB 2022/116.0411). Peripheral blood mononuclear cells (PBMCs) were collected from healthy donors (written informed consents were obtained from all participants) and isolated using Lymphoprep™ (Alere Technologies AS, Oslo, Norway) following manufacturer's protocol. In brief, the blood was diluted in ratio 1:1 with Phosphate buffer saline (PBS) and gently layered on top of Lymphoprep™ solution. The separation was examined by centrifugation at 800 $g$ for 30 min without decline acceleration then PBMCs layer was collected and washed twice with RPMI-1640 medium (*Chatupheeraphat et al., 2020*).

### Measurement of cell cytotoxicity by MTT assay

Leukemic cell lines ($1.5 \times 10^4$ cells/mL) and normal PBMCs ($1.0 \times 10^5$ cells/mL) were treated with various concentration of menthol (0, 100, 200, 300 μg/mL) and/or daunorubicin (0.00, 0.25, 0.50, 0.75, 1.00 μg/mL) in 96-well plate and incubated for 24 and 48 h, respectively, at 37 °C with 5% $CO_2$. After that, 10 μl of 5 mg/mL MTT solution was added and incubated for 4 h at 37 °C. The formazan crystal was solubilized by adding 100 μl of 10% SDS in 0.01 M HCl and further incubation for overnight. The absorbance of formazan was measured using microplate reader at 570 nm with Gen5™ analysis software (BioTek Instruments, Inc., Winooski, VT, USA). Then, the half maximal inhibitory concentration ($IC_{50}$) was calculated for further experiments.

Leukemic cells and PBMCs were treated with $IC_{50}$ concentration at 48 h of menthol and/or daunorubicin (TOKU-E, Bellingham, WA, USA), which are 250 and 0.2 μg/mL, respectively, to examine the synergistic property of menthol and daunorubicin. Cell viability was measured by MTT assay, whereas combination index (CI) was used to analyze

drug interactions following the equation: $Combination\ Index\ (CI) = \frac{(D)1}{(Dx)1} + \frac{(D)2}{(Dx)2}$

where $(D)1$ and $(D)2$ are concentrations of the first compound and second compound that achieve X% inhibition in the combination; $(Dx)1$ and $(Dx)2$ are the concentration of the first compound (alone) and second compound (alone), respectively, that represent the same effect (*Subkorn et al., 2021*).

## Apoptosis assay by annexin V/PI staining

In brief, leukemic cell line ($1.0 \times 10^5$ cells/mL) were treated with $IC_{50}$ concentration (250 µg/mL) of menthol and incubated for 48 h at 37 °C with 5% $CO_2$. Then, leukemic cells were collected and washed with PBS, stained with 3 µl of both Annexin V-FITC and 50 ug/mL propidium iodide (PI), respectively. FACSCantoII flow cytometer was used to examine the apoptotic cells, which analyzed as quantitative results by using FACSDiva software (BD bioscience, Palo Alto, CA, USA) (*Chatupheeraphat et al., 2020*).

## Determination of LC3-II autophagic marker by flow cytometry

Briefly, leukemic cell line ($1.0 \times 10^5$ cells/mL) were treated with $IC_{50}$ concentration (250 µg/mL) of menthol and incubated for 48 h at 37 °C with 5% $CO_2$. Before the end of incubation time, diluted autophagy reagent A as a lysosomal inhibitor was added, then the cells were incubated for 30 min. After that, the cell pellets were resuspended in 100 µl autophagy reagent B and stained with FITC-conjugated anti-LC3-II antibodies for 30 min. The LC3-II level referred as quantitative autophagosomes was determined by flow cytometry.

## Gene expression by reverse transcription quantitative PCR (RT-qPCR)

Leukemic cell lines were treated with $IC_{50}$ concentration of menthol for 48 h at 37 °C with 5% $CO_2$, then total RNA was extracted using Genezol™ reagent (New England Biolab, Inc., Ipswich, MA, USA) following manufacturer's protocol. Nanodrop2000 (Thermo Scientific, Waltham, MA, USA) was used to determine the RNA concentration. Then, 2 µg of total RNA was converted into cDNA by RevertAid first stand cDNA synthesis kit (Thermo Scientific, Waltham, MA, USA). qPCR was performed using the designed primers as shown in Table 1. cDNA was amplified by Luna® real time PCR master mix (New England Biolab, Inc., Ipswich, MA, USA) in Bio-Rad CFX96 touch™ real time PCR system (Bio-Rad, Hercules, CA, USA). The mRNA expression was analyzed by $2^{-\Delta\Delta CT}$ method using GAPDH as an internal control.

## Prediction of protein-chemical interactions using bioinformatic tools

The targeted proteins of menthol were constructed by STITCH database (http://stitch.embl.de/), which is computational tool for prediction of molecule and its protein interaction using any sources from previous experiments. Menthol was used as a keyword for search parameter with apoptotic and autophagic proteins. The medium confidence (0.400) was considered for analysis of menthol and its protein interaction.

**Table 1 The primers used in RT-qPCR.**

| Primers | Primer sequence (5′-3′) |
| --- | --- |
| Caspase-3 | F: 5′-TTCAGAGGGGATCGTTGTAGAAGTC-3′ |
| | R: 5′-CAAGCTTGTCGGCATACTGTTTCAG-3′ |
| BAX | F: 5′-CGAGAGGTCTTTTTCCGAGTG-3′ |
| | R: 5′-GTGGGCGTCCCAAAGTAGG-3′ |
| p53 | F: 5′-CCCAGCCAAAGAAGAAACCA-3′ |
| | R: 5′-TCTGAGTCAGGCCCTTCTGT-3′ |
| MDM2 | F: 5′-CAGAGTCTTGCTCCATCACC-3′ |
| | R: 5′-ATGCCTGTAATCCCAGTTACTTG-3′ |
| mTOR | F: 5′-CGCTGTCATCCCTTTATCG-3′ |
| | R: 5′-ATGCTCAAACACCTCCACC-3′ |
| ATG3 | F: 5′-CACGACTATGGTTGTTTGGCTATG-3′ |
| | R: 3′-GGTGGAAGGTGAGGGTGATTT-5′ |
| GAPDH | F: 5′-GCACCGTCAAGGCTGAGAA-3′ |
| | R: 5′-AGGTCCACCACTGACACGTTG-3′ |

## Determination of protein expression by western blot analysis

The expression levels of apoptosis-related proteins and proteins involved in autophagy were analyzed by western blot. Leukemic cells were treated with $IC_{50}$ concentration of menthol for 48 h and harvested for protein extraction, which extracted by using RIPA lysis buffer then incubated on ice for 30 min and centrifuged at 14,000 $g$ for 10 min at 4 °C to collect the suspension. Extracted proteins were separated by sodium dodecyl sulfate-polyacrylamide gel electrophoresis (SDS-PAGE) and blotted onto polyvinylidene fluoride membranes for 2 h. After that, the membranes were blocked with 5% non-fat milk and/or 5% w/v BSA in Tris-buffered saline containing 0.1% Tween 20 (pH 7.4) at room temperature for 2 h and incubated with primary antibodies (Cell signaling, Danvers, MA, USA) for caspase-3 (RRID:AB 2341188), mTOR (RRID:AB 2105622), and β-actin (RRID: AB 2242334) overnight. Blots were incubated with horseradish peroxidase (HRP)-conjugated secondary antibodies, including anti-mouse IgG, HRP-linked antibody (Cell signaling, Massachusetts, USA, RRID:AB 330924) and Goat anti-Rabbit IgG antibody, (H+L) HRP conjugate (Millipore, Darmstadt, Germany, RRID:AB 92641). After that, the signals were detected *via* enhanced chemiluminescence.

## Statistical analysis

All experiments were conducted in triplicate and all results were represented in mean ± SEM. GraphPad Prism8 (GraphPad Inc., San Diego, CA, USA) was used in all statistical analyses including student's t-test for two group comparison and one-way ANOVA for more than two group comparison. $P$-value < 0.05 was considered as a statistically significant difference.

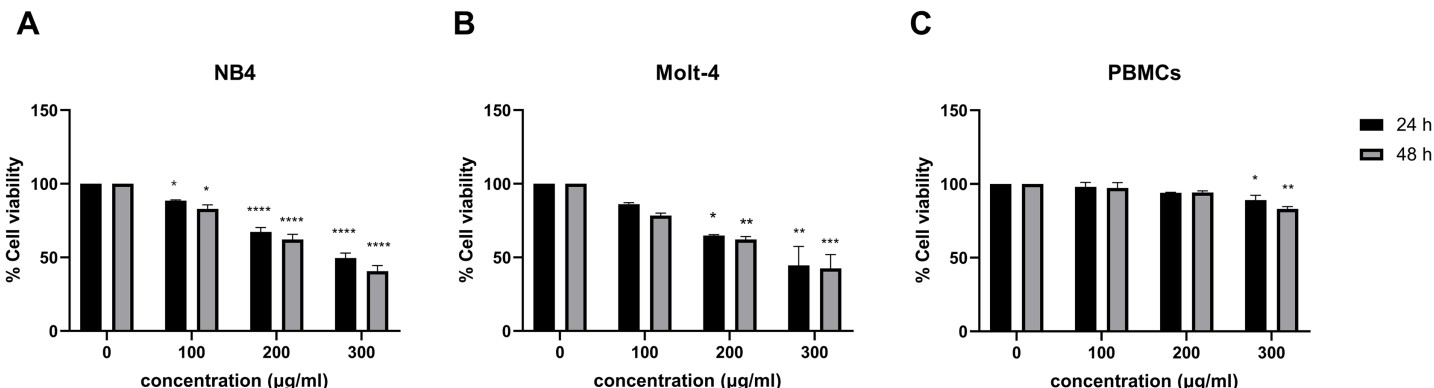

**Figure 1 Effect of menthol on cell viability in leukemic cells and PBMCs.** (A) NB4, (B) Molt-4 and (C) PBMCs were treated with various concentrations (0, 100, 200 and 300 μg/ml) of menthol for 24 and 48 h. MTT assay was used to assess cell viability. $*p < 0.05$, $**p < 0.01$, $***p < 0.001$, and $****p < 0.0001$ were considered as a statistically significant difference from control groups.

## RESULTS

### Cytotoxicity effect of menthol on leukemic cell lines

The effect of menthol on cell viability was observed in menthol treated leukemic cell lines using MTT assay. Figures 1A and 1B, menthol significantly decreased cell viability in a dose-dependent manner at both 24 and 48 h in NB4 and Molt-4 cell lines. The comparative results of menthol-treated group and control group were shown in Table 2. The $IC_{50}$ concentrations of menthol were 296.7 and 250.9 μg/mL in NB4 cells while they were 280.7 and 247.5 μg/mL in Molt-4 cells after 24 and 48 h of treatment, respectively. After that, the $IC_{50}$ at 48 h in both NB4 and Molt-4 cells was used for further experiments. However, a minimal cytotoxicity was observed in normal PBMCs as shown in Fig. 1C.

### Effect of combination between menthol and daunorubicin on leukemic cells

Leukemic cells were treated with $IC_{50}$ concentration of menthol and/or daunorubicin for 48 h compared with normal PBMC, then cell viability was assessed using MTT assay. Cell viability of daunorubicin treated leukemic cells was represented in Fig. 2 and comparative results represented in Table 3. The results showed that combined menthol with daunorubicin significantly decreased the percentage of cell viability when compared to menthol or daunorubicin treatment alone in both NB4 ($p = 0.0001$ and 0.0002, respectively) and Molt-4 leukemic cells ($p = 0.0001$ and 0.005, respectively) (Figs. 3A and 3B), whereas cytotoxicity of PBMCs was not affected (Fig. 3C). Moreover, the combination index (CI) was calculated for drug interaction (*i.e.*, menthol and daunorubicin), which demonstrated a significant synergistic effect on both leukemic cells with CI value of 0.79 and 0.94 for NB4 and Molt-4, respectively. The combination index (CI) values were referred a drug interaction as synergism (CI < 1), additive (CI = 1), and antagonism (CI > 1).

**Table 2 The comparative results of the menthol-treated group and the control group.**

| Group | Conc. (µg/ml) | % Cell viability of NB4 (mean ± S.E.M) | p-value | Group | Conc. (µg/ml) | % Cell viability of Molt-4 (mean ± S.E.M) | p-value |
|---|---|---|---|---|---|---|---|
| 24 h | | | | | | | |
| Control | 0 | 100 ± 0.00 | – | Control | 0 | 100 ± 0.00 | – |
| Menthol | 100 | 88.53 ± 0.53 | 0.027177 | Menthol | 100 | 86.20 ± 1.01 | 0.472171 |
| | 200 | 67.34 ± 2.92 | 0.000030 | | 200 | 64.79 ± 0.62 | 0.019908 |
| | 300 | 49.61 ± 3.33 | 0.000001 | | 300 | 44.61 ± 12.80 | 0.001326 |
| $IC_{50}$ = 296.7 µg/ml | | | | $IC_{50}$ = 270.5 µg/ml | | | |
| 48 h | | | | | | | |
| Control | 0 | 100 ± 0.00 | – | Control | 0 | 100 ± 0.00 | – |
| Menthol | 100 | 82.99 ± 2.72 | 0.013312 | Menthol | 100 | 78.50 ± 1.58 | 0.057148 |
| | 200 | 62.18 ± 3.47 | 0.000069 | | 200 | 62.25 ± 1.96 | 0.002689 |
| | 300 | 40.68 ± 3.73 | 0.000002 | | 300 | 42.54 ± 9.46 | 0.000154 |
| $IC_{50}$ = 250.9 µg/ml | | | | $IC_{50}$ = 257.6 µg/ml | | | |

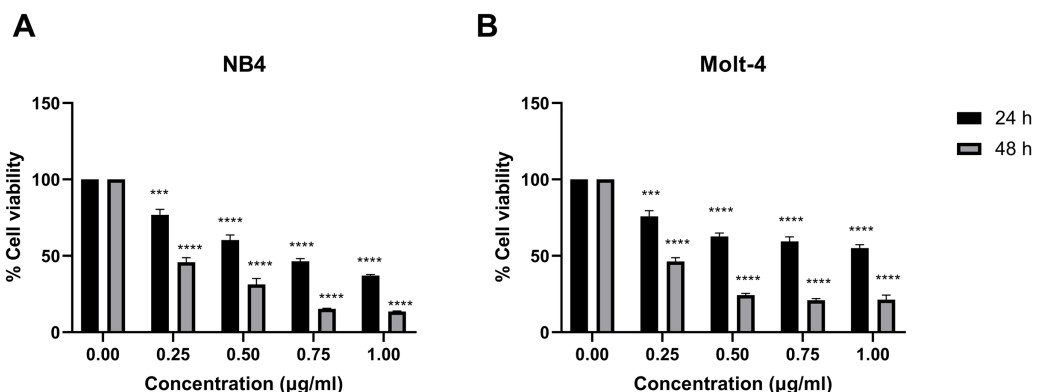

**Figure 2 Effect of daunorubicin on cell viability in NB4 and Molt-4 leukemic cells.** (A) NB4 and (B) Molt-4 were treated with various concentrations (0, 0.25, 0.50, 0.75 and 1.00 µg/ml) of daunorubicin for 24 and 48 h. Cell viability were measured using MTT assay. ***$p < 0.001$, and ****$p < 0.0001$ were considered as a statistically significant difference from control groups.

## Menthol induces apoptosis in leukemic cell lines

Leukemic cells were treated with menthol with $IC_{50}$ concentration for 48 h, then cell apoptosis was analyzed by flow cytometry with annexin V-FITC staining. As shown in Fig. 4, menthol increased the percentage of apoptotic cells which includes early (Q2) and late apoptosis (Q4) in both NB4 and Molt-4 with 44.86 ± 4.89% and 51.36 ± 0.57% ($p$ = 0.002 and 0.00005, respectively). Therefore, the results referred that menthol significantly induced apoptosis in NB4 and Molt-4 leukemic cells.

## Menthol increases LC3-II levels for autophagy induction

NB4 and Molt-4 leukemic cells were treated with $IC_{50}$ concentration for 48 h to evaluate the induction of autophagy analyzed by flow cytometry. The levels of LC3-II, which is a

Table 3 The comparative results of the daunorubicin-treated group and the control group.

| Group | Conc. (µg/ml) | % Cell viability of NB4 (mean ± S.E.M) | p-value | Group | Conc. (µg/ml) | % Cell viability of Molt-4 (mean ± S.E.M) | p-value |
|---|---|---|---|---|---|---|---|
| 24 h | | | | | | | |
| Control | 0.00 | 100 ± 0.00 | – | Control | 0.00 | 100 ± 0.00 | – |
| daunorubicin | 0.25 | 76.86 ± 2.53 | 0.000272 | daunorubicin | 0.25 | 45.79 ± 2.96 | 0.000436 |
| | 0.50 | 60.32 ± 3.35 | 0.000002 | | 0.50 | 31.35 ± 3.82 | 0.000010 |
| | 0.75 | 46.50 ± 1.72 | 0.0000002 | | 0.75 | 15.28 ± 0.35 | 0.000005 |
| | 1.00 | 37.04 ± 0.74 | 0.00000002 | | 1.00 | 13.51 ± 0.43 | 0.000002 |
| $IC_{50}$ = 0.7 µg/ml | | | | $IC_{50}$ = 1.3 µg/ml | | | |
| 48 h | | | | | | | |
| Control | 0.00 | 100 ± 0.00 | – | Control | 0.00 | 100 ± 0.00 | – |
| daunorubicin | 0.25 | 75.76 ± 3.76 | 0.00000006 | daunorubicin | 0.25 | 46.26 ± 1.578 | 0.00000002 |
| | 0.50 | 62.72 ± 2.17 | 0.000000004 | | 0.50 | 24.36 ± 1.960 | 0.0000000004 |
| | 0.75 | 59.41 ± 3.07 | 0.0000000005 | | 0.75 | 20.90 ± 9.457 | 0.0000000003 |
| | 1.00 | 55.06 ± 2.27 | 0.0000000004 | | 1.00 | 21.30 ± 3.329 | 0.0000000003 |
| $IC_{50}$ = 0.2 µg/ml | | | | $IC_{50}$ = 0.2 µg/ml | | | |

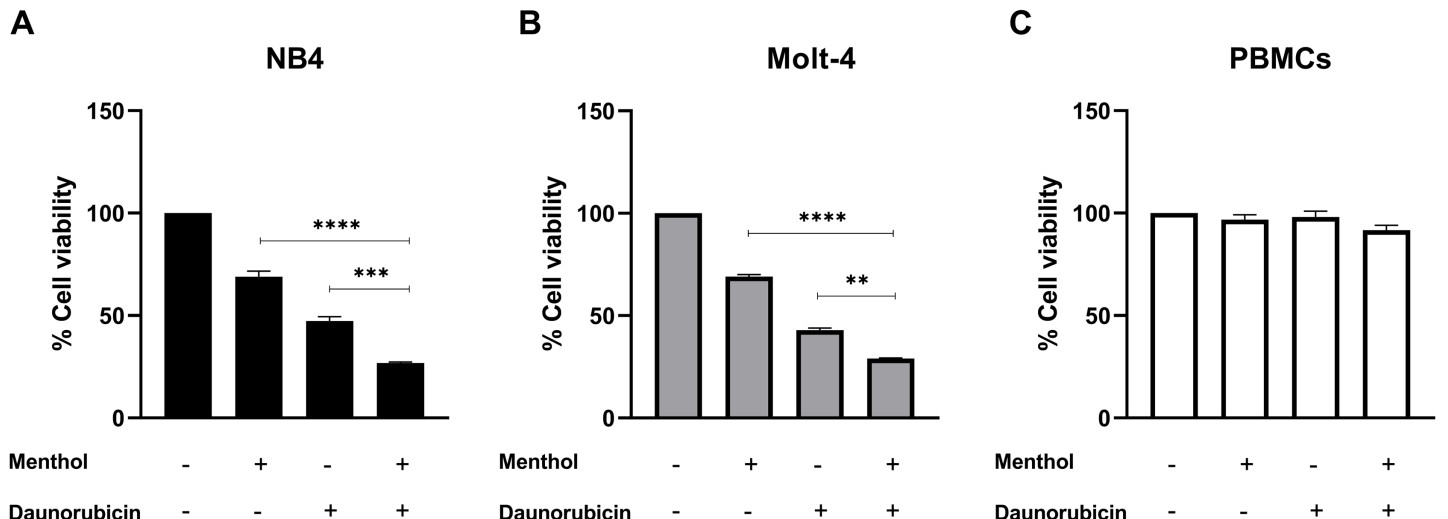

Figure 3 The synergistic effect induced by combination of menthol and daunorubicin on leukemic cells and PBMCs. (A) NB4, (B) Molt-4 and (C) PBMCs were treated with $IC_{50}$ concentrations of menthol and daunorubicin for 48 h. Cell viability was assessed using MTT assay. $**p < 0.01$, $***p < 0.001$, and $****p < 0.0001$ were considered as a statistically significant difference from control groups.

marker for autophagic cells were measured and analyzed. The results indicated that menthol significantly induced levels of LC3-II in NB4 and Molt-4 leukemic cells when compared to the control as represented in Fig. 5. ($p = 0.006$ and 0.002, respectively).

## Protein-menthol interactions predicted by STITCH

STITCH database was used to predict the interplay between menthol and proposed proteins, which are CASP3, BAX, ATG3, MTOR, TP53, and MDM2. The results from

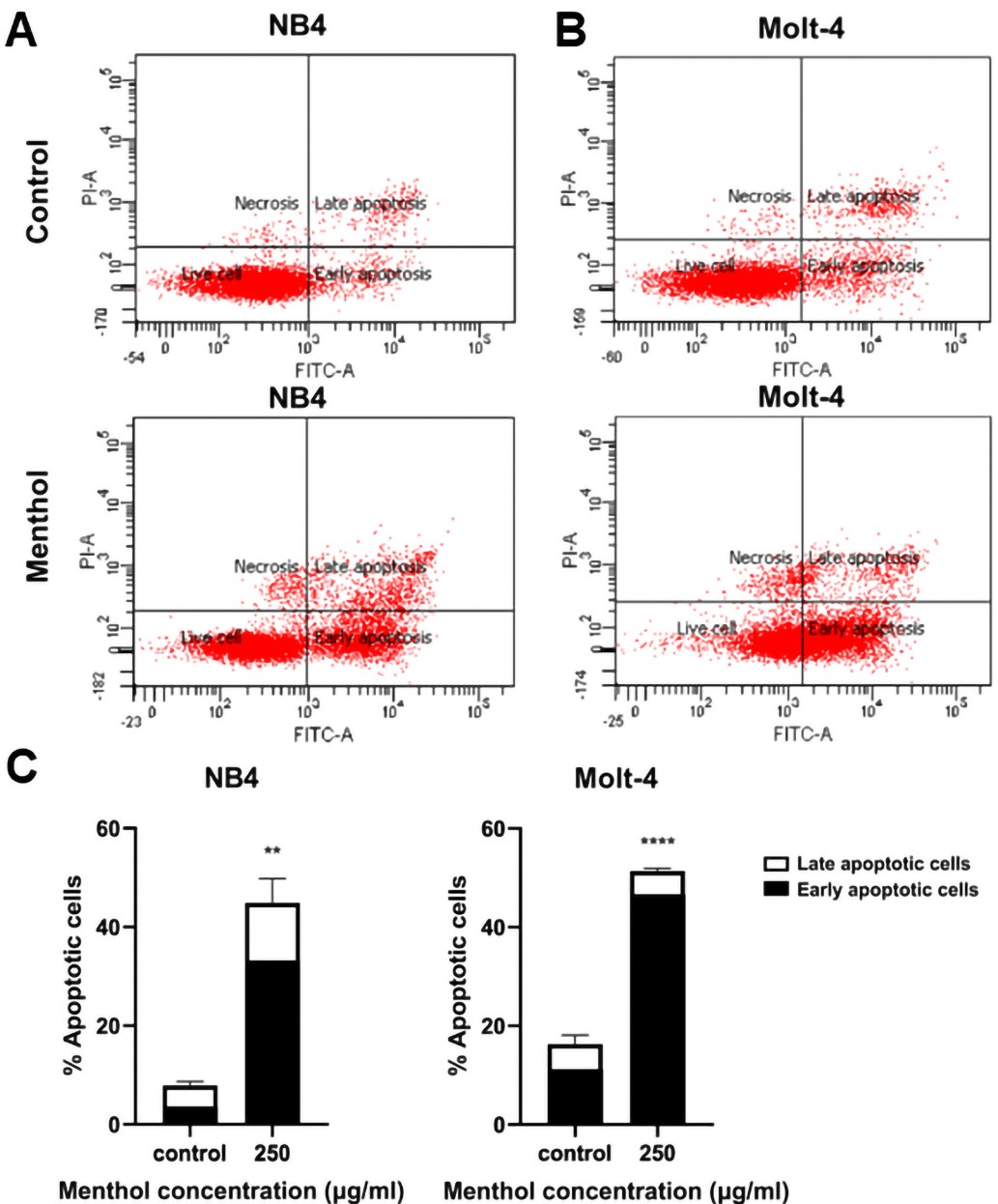

**Figure 4 Effect of menthol on apoptosis induction in NB4 and Molt-4 leukemic cells.** NB4 and Molt-4 were treated with menthol at $IC_{50}$ concentration for 48 h and apoptotic cells were measured by using flow cytometry with annexin V-FITC/PI staining. Scatter plots of (A) NB4 and (B) Molt-4 were shown the menthol compared to control. Early and late apoptosis of NB4 and Molt-4 leukemic cells were shown in (C) quantitative result. $^{**}p < 0.01$, and $^{****}p < 0.0001$ were considered as a statistically significant difference from control groups.

STITCH software analysis found that menthol was directly associated with CASP3 (0.700), which represented high-score interactions with other proposed proteins including BAX (0.869), MTOR (0.845), TP53 (0.914), and MDM2 (0.994). ATG3 was also directly associated with menthol in a high score combination (0.530), which related to MTOR (0.555) and ULK1 (0.886) as shown in Fig. 6. Moreover, RPTOR, RICTOR, RPS6KB1,

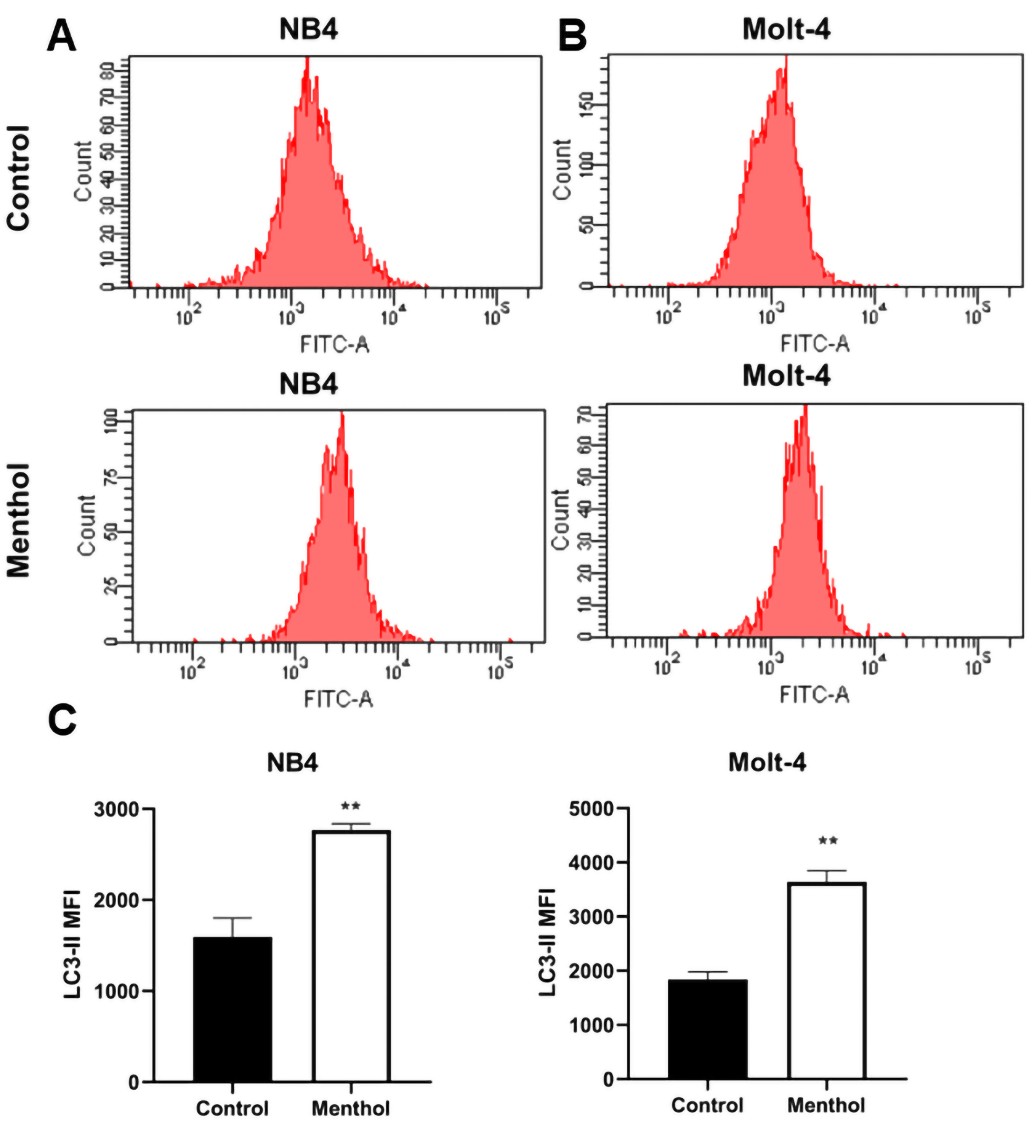

**Figure 5 Effect of menthol on autophagy induction by increasing LC3-II levels.** (A) NB4 and (B) Molt-4 were treated with $IC_{50}$ concentration of menthol for 48 h and analyzed LC3-II as the mean fluorescence intensity (MFI) by flow cytometry. (C) Quantitative result of LC3-II levels in both leukemic cells compared control. $^{**}p < 0.01$, was considered as a statistically significant difference from control groups.

MLST8, AKT1, EIF4EBP1, FKBP1A, and MAPKAP1 proteins were related to MTOR at high-score combinations (0.999), whereas CDKN1A and ATM showed a high-score linkage with CASP3 (0.992) and TP53 (0.999), respectively.

## Effects of menthol on apoptotic and autophagic mRNA expressions

Pro-apoptotic mRNA expression, including CASP3, BAX, TP53 and autophagic mRNA expression including ATG3 and MTOR were measured by RT-PCR. The results showed

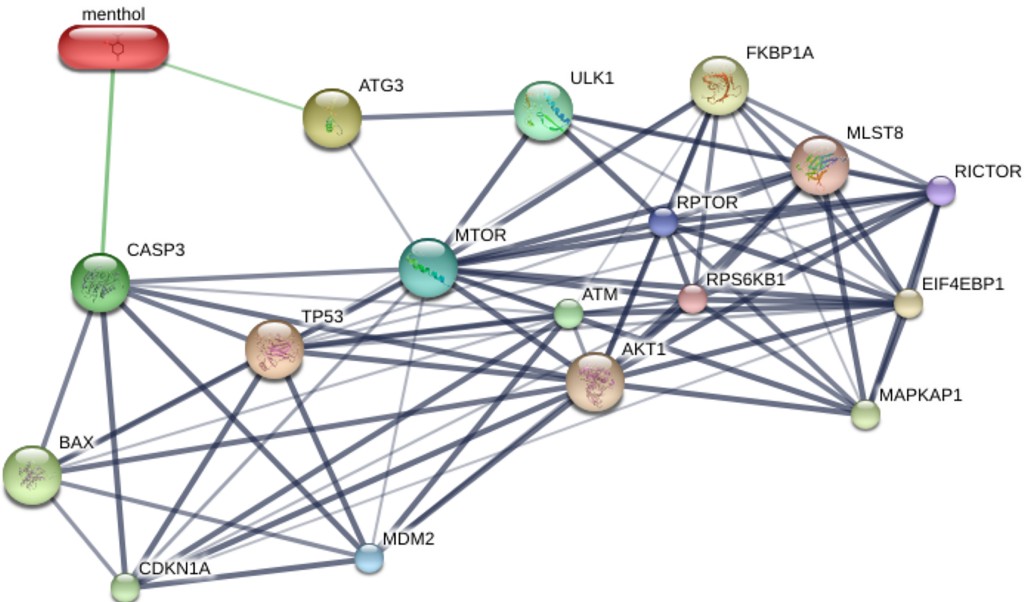

**Figure 6 Construction of chemical and proposed protein interaction using STITCH database.** Interactions between menthol and proteins are represented by green lines, while protein interactions are demonstrated by grey lines. Thicker lines are referred a strong association. CASP3, caspase-3; BAX, BCL-2 associated X protein; ATG3, autophagy related 3; MTOR, mechanistic target of rapamycin; TP53, tumor protein p53; MDM2, Mdm2; ULK1, unc-51-like kinase 1; RPTOR, regulatory associated protein of MTOR; RICTOR, RPTOR independent companion of MTOR; RPS6KB1, ribosomal protein S6 kinase; MLST8, MTOR associated protein; AKT1, v-akt murine thymoma viral oncogene homolog 1; EIF4EBP1, eukaryotic translation initiation factor 4E binding protein; FKBP1A, FK506 binding protein 1A; MAPKAP1, mitogen-activated protein kinase associated protein 1; CDKN1A, cyclin-dependent kinase inhibitor 1A; ATM, ataxia telangiectasia mutated.

that Caspase-3 ($p = 0.00008$ for NB4; $p = 0.0009$ for Molt-4), BAX ($p = 0.001$ for NB4; $p = 0.0001$ for Molt-4), p53 ($p = 0.000002$ for NB4; $p = 0.000000001$ for Molt-4) was significantly upregulated by menthol treatment in both NB4 and Molt-4 leukemic cells. In contrast, anti-apoptotic gene, which is MDM2 as a negative regulator of TP53 was significantly downregulated in both leukemic cells ($p = 0.0001$ for NB4; $p = 0.0002$ for Molt-4). Moreover, menthol also significantly increased mRNA expression of ATG3 ($p = 0.007$ for NB4; $p = 0.00007$ for Molt-4) and decreased MTOR ($p = 0.01$ for NB4; $p = 0.00009$ for Molt-4) in both NB4 and Molt-4 leukemic cells (Fig. 7).

## Effects of menthol on menthol-related apoptotic and autophagic proteins

Western blot analysis was performed for studying level of proteins related apoptosis and autophagy, including Caspase-3 ($p = 0.002$ for NB4; $p = 0.0009$ for Molt-4) and mTOR ($p = 0.003$ for NB4; $p = 0.001$ for Molt-4), respectively. The results revealed that menthol significantly upregulated the levels of CASP3 and downregulated MTOR protein expression in both leukemic cells when compared to the control group (Fig. 8).

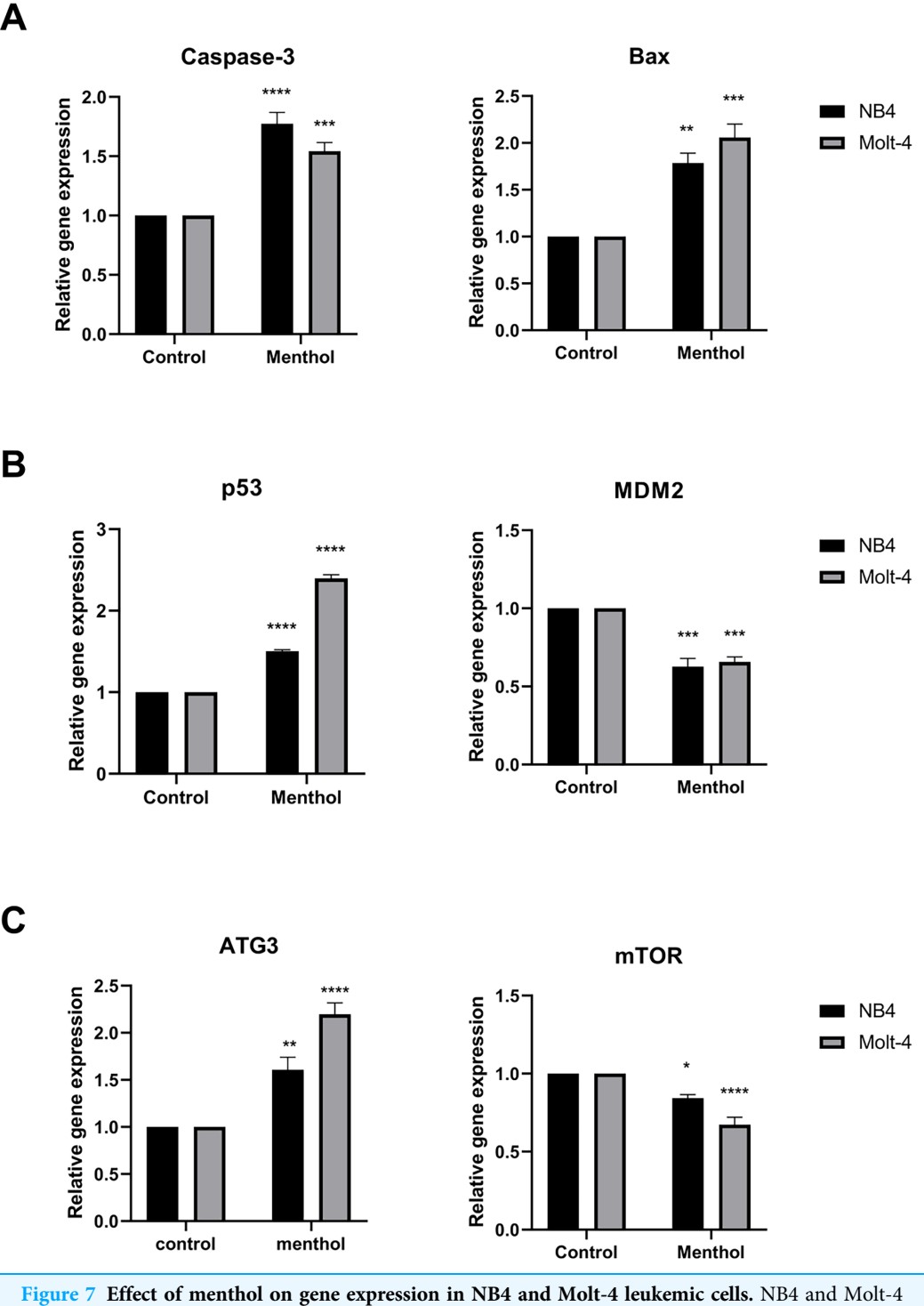

**Figure 7 Effect of menthol on gene expression in NB4 and Molt-4 leukemic cells.** NB4 and Molt-4 were treated with menthol at $IC_{50}$ concentration for 48 h and analyzed relative gene expression levels using RT-PCR. Quantitative result of (A) Caspase-3 and BAX, (B) p53 and MDM2, and (C) ATG3 and mTOR were compared to control. $^*p < 0.05$, $^{**}p < 0.01$, $^{***}p < 0.001$, and $^{****}p < 0.0001$ were considered as a statistically significant difference from control groups.

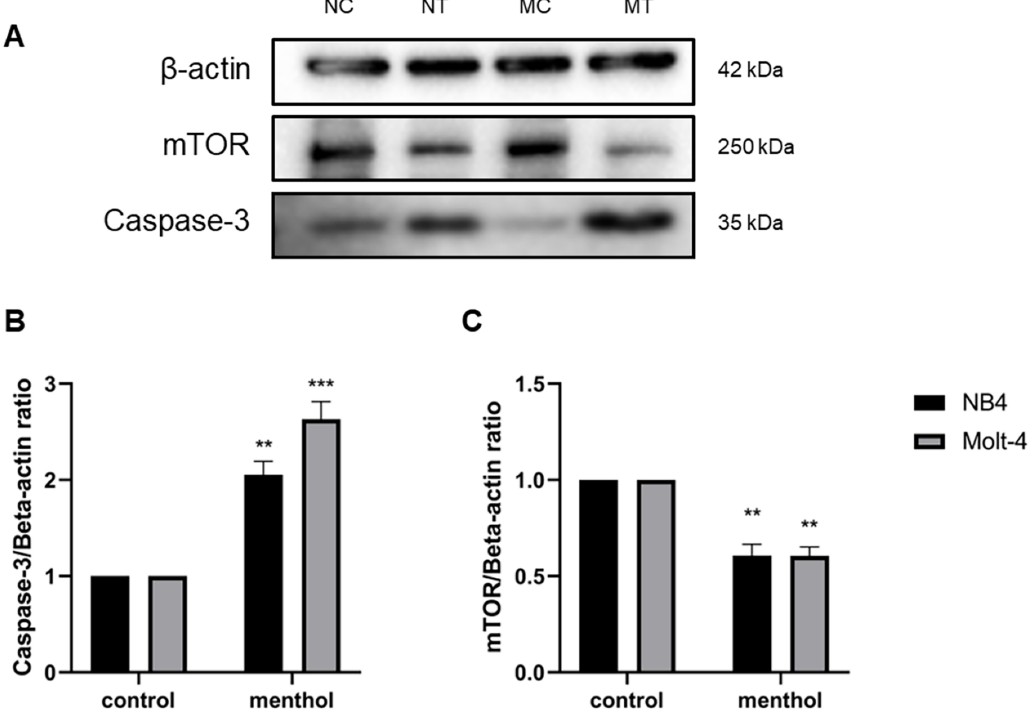

**Figure 8 Effect of menthol on protein expression in NB4 and Molt-4 leukemic cells.** NB4 and Molt-4 were treated with menthol at $IC_{50}$ concentration for 48 h and western blot was used to determine ratio between targeted protein and β-actin as internal control (A) Protein expression of (B) Caspase-3 and (C) mTOR in both leukemic cells were represented in quantitative result. NC, NB4 control; NT, NB4 treated; MC, Molt-4 control; MT, Molt-4 treated. $^{**}p < 0.01$ and $^{***}p < 0.001$ were considered as a statistically significant difference from control groups.

## DISCUSSION

Leukemia is one concerned health problems worldwide, even the treatment approaches have developed, in contrast the disease relapsing and other side effects associated chemotherapy drugs are a great issue. The advances in leukemia treatment have significantly improved the survival rate of patients, however, the survival rates still depend on several factors that relate to the tolerance against the adverse effects in patients (*Chatupheeraphat et al., 2020*). Moreover, there are approximately 50% of leukemia patients in both myeloid and lymphoid leukemic types who experience a disease relapse after initial treatment. Thus, to reduce the possible side effects related chemotherapy treatment, the alternative medicines using natural compounds have been focused on recent research studies.

In this study, we investigated the anti-leukemic effects of menthol on NB4 and Molt-4 cell lines, which represent as myeloid and lymphoid leukemia, respectively. Menthol is an organic compound found most in peppermint leaves that exhibits several biological properties such as antioxidant, antiviral, antimicrobial, antifungal, and anticancer (*Almatroodi et al., 2021*; *Fatima et al., 2021*; *Kim et al., 2012*; *Li et al., 2009b*; *Lu et al., 2006*; *Nagai et al., 2019*). The results revealed that menthol induced cytotoxicity by decreasing

cell viability of both NB4 and Molt-4 leukemic cells in a dose-dependent manner and exerted a minimal effect on PBMCs. In addition, menthol enhanced apoptosis by increasing the levels of apoptotic cells using annexin V-FITC staining in both leukemic cells. Autophagy was also examined to be exerted by menthol in both NB4 and Molt-4 by increasing level of LC3-II. These results related to the gene and protein expressions associated with apoptosis and autophagy using RT-PCR and western blot, which shown that menthol increased mRNA expression of pro-apoptotic genes (*i.e.*, Caspase-3, BAX, p53), reduced anti-apoptosis gene (*i.e.*, MDM2). For autophagic gene, ATG3 as an activator and mTOR as an inhibitor was upregulated and downregulated after menthol treatment, respectively. Concomitant with the protein expressions, Caspase-3 and mTOR were upregulated and downregulated by menthol, respectively in both NB4 and Molt-4 leukemic cells.

In previous studies, menthol has reported as anti-cancer agents in several cancer types *via* the transient receptor potential melastatin 8 (TRPM8)-dependent pathway or in TRPM8-independent manner. For example, the cell death induction in human bladder cancer T24 cells has associated with TRPM8, which occurs by menthol increased the concentration of intracellular calcium and induced mitochondrial membrane depolarization through TRPM8 channels (*Li et al., 2009b*). In addition, menthol has exerted the anti-tumor activity in prostate cancer DU145 cells *via* upregulating the TRPM8 expression by menthol, which leads to anti-proliferation and inhibition of motility. The results have revealed that menthol is able to induce cell cycle arrest at $G_0/G_1$ phase and downregulate the focal-adhesion kinase (FAK) pathway, which is involved in cell migration, cell proliferation, and cell survival (*Wang et al., 2012*). In leukemia, there are less of evidence for menthol induced cell death through the death signaling pathways, however, *Lu et al. (2006)* has studied the effects of menthol on human promyelocytic leukemia HL-60 cell death. The results exhibit that menthol can enhance HL-60 cell death through necrosis, not exerting in apoptosis and no cell cycle arrest has found. Moreover, menthol induces the production of $Ca^{2+}$ leading to the induction of menthol-induced HL-60 cell death (*Lu et al., 2006*).

Using the natural products combined with chemotherapy drugs has shown the improvement in cancer treatment, which are reducing the chemotherapy induced toxic effects, decreasing dose of therapeutic drugs, and suppressing the development of drug resistance (*Deesrisak et al., 2021b*). The combination has the potential to be an effective drug with maximize treatment efficacy whereas minimize the potential adverse effects (*Sauter, 2020*). For example, curcumin combined with a chemotherapy medicine, paclitaxel have a potential to inhibit MDA-MB-231 and MCF-7 breast cancer cell growth and induce the apoptosis signaling pathway through upregulating Caspase-3, Caspase-8, BAX, p53, oxidative stress (*Kang et al., 2009*; *Quispe-Soto & Calaf, 2016*; *Zhan et al., 2014*; *Zhang et al., 2020*). In this study, the combination of menthol and daunorubicin was determined. Menthol was shown the potential to enhances the cytotoxicity of daunorubicin with less toxicity in PBMCs. However, the mechanism of actions of menthol combined daunorubicin is still unknown. Daunorubicin is chemotherapy medicine used in cancer treatment, especially leukemia. Its actions caused by DNA intercalation and

inhibition of DNA synthesis, in addition, daunorubicin induces reactive oxygen species (ROS) leading to DNA damaged and apoptosis (*Al-Aamri et al., 2019*; *Pourahmad, Salimi & Seydi, 2016*).

Nowadays, previous studies have focused on apoptosis and autophagy as main targeted mechanisms for cancer therapy. Mostly in cancer, the apoptosis pathway is typically inhibited by overexpression of antiapoptotic proteins. Therefore, the promising anticancer agents that exhibit anticancer activity through enhancing apoptosis are focused (*Pfeffer & Singh, 2018*). For example, linalool, a plant-derived monoterpene, induces $G_0/G_1$ arrest and apoptosis in hepatocellular carcinoma HepG2 cells through Ras, MAPKs, Akt/mTOR signaling pathways (*Rodenak-Kladniew et al., 2018*). Additionally, piperine, a natural compound of black peppers, inhibits human melanoma cell growth and induces apoptosis *via* Caspase/BAX signaling proteins, while suppresses antiapoptotic expression and phosphor-ERK1/2 proteins (*Yoo et al., 2019*). On the other hand, autophagy in cancer cells may exert dual roles, including contribute the antitumor effects and enhance the tumor progression; however, several anticancer drugs induce autophagy *via* the PI3K/Akt/mTOR signaling pathway (*Janku et al., 2011*). Consequently, the induction of autophagic cancer cells by natural products is commonly used for development of novel therapeutic agents. For example, sesamin, a lignan isolated from bark of Fagara plants, is able to enhance ER stress-mediated apoptosis *via* IRE1α/JNK pathways, which activate autophagy induction resulting the autophagic death in cervical cancer (HeLa) cells (*Dou et al., 2018*). Ailanthone and diallyl disulfide, which are the natural compounds from Chinese plants and garlic, respectively, induce autophagy through Beclin1, p62, LC3 expression (*Wei et al., 2018*) and mTOR-mediated autophagic death in myeloid leukemia cells (*Suangtamai & Tanyong, 2016*). Moreover, many researchers have studied for the better understanding the crosstalk between apoptosis and autophagy. Since, there are several factors modulate the two pathways, *Qian et al. (2017)* reveled the proposed mechanisms underlying the relationship among apoptosis and autophagy. Firstly, apoptosis and autophagy can be a concomitantly pathway, while regulate cell death independently, secondly, one leads to the other in both apoptosis and autophagy. Thirdly, autophagy prevents accumulation of damaged DNA and ER stress products resulting in apoptosis inhibition (*Li et al., 2009a*; *Qian et al., 2017*; *Rikiishi, 2012*; *Takahashi et al., 2014*). Therefore, this present study demonstrated that menthol enhanced apoptosis through the Caspase and BAX activation, p53/MDM2 pathway, moreover, induce autophagy induction by increasing LC3-II levels and ATG3/mTOR-mediated autophagic death in human leukemic cells.

Bioinformatics tools, which are the programs for prediction using available databases of biological data, which are useful for understanding of pharmacology and biochemistry. To develop therapeutic drugs, prediction of protein and small molecule interactions is a great of interest for finding, improving, and developing the drugs for diseases (*Xia, 2017*). The interaction between menthol and proposed associated proteins (Caspase-3, BAX, p53, MDM2, ATG3 and mTOR) revealed that many targets involved in the response to menthol, including RICTOR, RRTOR, MLST8, RPS6KB1 and AKT1, which play a role in mTOR signaling pathway, have reported their overexpression in several cancer cells such as breast, lung, colon cancer cells. Downregulation of these proteins associated with

inhibition of tumor cell growth and enhancement of cancer cell death (*Bahrami-B et al., 2014*; *Gulhati et al., 2009*; *Kakumoto et al., 2015*; *Malla et al., 2015*; *Martin et al., 2016*). The other associated proteins, translation factors EIF4EBP1, CDKN1A, and MAPKAP1 are related to leukemogenesis, a regulator of cell cycle, and a kinase protein involved in PI3K/MAPK pathway respectively (*Kovalski et al., 2019*; *Kreis, Louwen & Yuan, 2019*; *Topisirovic et al., 2003*). In addition, FKBP1A has its roles in tumor progression and metastasis (*Patel et al., 2022*).

In a present study suggested that menthol could be an effective anti-leukemic agent, which exerted activities for cell viability inhibition in both NB4 and Molt-4 leukemic cells. These effects were associated with regulation of apoptosis signaling pathway and the induction of autophagy. The expressions of important apoptotic and autophagic genes in both activator (*i.e.*, Caspase-3, BAX, p53, ATG3) and inhibitor factors (*i.e.*, MDM2, mTOR) was upregulated and downregulated in NB4 and Molt-4 cells, respectively. These findings suggested the potent anti-leukemic effects of menthol on NB4 and Molt-4 cells *via* the apoptosis and autophagy signaling pathways.

# CONCLUSIONS

The findings in this present study revealed that menthol exhibits anti-leukemic activities by inhibiting cell viability, inducing apoptosis *via* upregulated Caspase-3, BAX, p53 while downregulated MDM2, and enhancing autophagy through ATG3/mTOR in acute myeloid and lymphoid leukemic cell lines. However, further experiments are still needed for elucidation these finding. From the study significantly suggests of menthol that could be possible developed as a candidate therapeutic drug or combined drug for leukemia treatment in the future.

## Funding
This research project is supported by Mahidol University (Basic Research Fund: fiscal year 2022). The funders had no role in study design, data collection and analysis, decision to publish, or preparation of the manuscript.

## Grant Disclosures
The following grant information was disclosed by the authors:
Mahidol University.

## Competing Interests
The authors declare that they have no competing interests.

## Author Contributions
- Mashima Naksawat conceived and designed the experiments, performed the experiments, analyzed the data, prepared figures and/or tables, authored or reviewed drafts of the article, and approved the final draft.

- Chosita Norkaew performed the experiments, analyzed the data, prepared figures and/or tables, authored or reviewed drafts of the article, and approved the final draft.
- Kantorn Charoensedtasin analyzed the data, prepared figures and/or tables, authored or reviewed drafts of the article, and approved the final draft.
- Sittiruk Roytrakul conceived and designed the experiments, authored or reviewed drafts of the article, and approved the final draft.
- Dalina Tanyong conceived and designed the experiments, authored or reviewed drafts of the article, and approved the final draft.

## Ethics

The following information was supplied relating to ethical approvals (*i.e.*, approving body and any reference numbers):

The Mahidol University Central Institutional Review Board (MU-CIRB) approved the study (MU-CIRB 2022/116.0411).

## Data Availability

The raw measurement are available in the Supplemental Files.

## Supplemental Information

Supplemental information for this article can be found online at http://dx.doi.org/10.7717/peerj.15049#supplemental-information.

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
