# Peer review of "Anti-leukemic effect of menthol, a peppermint compound, on induction of apoptosis and autophagy"

_PeerJ, doi:10.7717/peerj.15049_

## Round 0.1 · original submission · Minor Revisions

All reviewers were positive about your study, and have made a series of suggestions which you should address. These fall into the category of minor changes, so I look forward to seeing the revised paper in due course. Please ensure you address all points raised.

Reviewer 1 ·

Basic reporting

The present study describes the anti-leukemic properties of menthol. In this study, NB4 and Molt-4 cells were used as a model for leukemia. The authors use a range of techniques to demonstrate their findings and show that menthol could be a promising therapeutic drug.

1. In lines 39, 341 and 351, the authors claim that menthol inhibits cell proliferation. In this study proliferation was not directly studied, instead, MTT assays were used to measure cell viability which should not be inferred as proliferation. Please provide evidence that menthol directly inhibits cell proliferation.
2. In lines 182/183 it is stated that “no cytotoxicity was observed in normal PBMCs”. However, in Figure 1C, a small but significant reduction in PBMC viability is shown. Please amend these lined to be in line with the figure and suggest a possible explanation for this observation.
3. In Figure 4, the authors demonstrate that menthol induces apoptosis in NB4 and Molt-4 cells. It would be interesting if you quantified the necrotic cells to determine if menthol also induces necrosis.
4. Please improve the clarity of the results shown in figure 5A. I suggest producing an overlay figure showing control and menthol-treated cells on the same panel.
5. At various points (lines 190, 194, 209, 224/225, among others) data is shown with up to 6 significant figures. For clarity this may be better represented as 2 significant figures.

Minor notes
1. In figure 1, there is no A/B/C on the figure. Please add these labels.
2. Please add a reference for line 45 for the details on cases/deaths of leukemia.
3. Please fix the formatting problem on lines 51-53.
4. In figure 4, please increase the font size of the axis and labels in figure A.

Experimental design

1. Please provide the catalogue details for the menthol used in this study. The supplier sells many types of menthol and this detail would help future researchers
2. Provide details of the primary antibodies used in this study. Please include the supplier and the RRID
3. Please provide details for the protein extraction technique mentioned in line 160
4. Please add the medium details to line 109

Validity of the findings

Figure 8B shows that menthol causes a significant increase in Caspase-3, however, the representative blot seems to indicate a decrease in Caspase-3. Please improve the clarity of this figure or provide an alternative blot for 8A.

Reviewer 2 ·

Basic reporting

I strongly advise the authors to revise thoroughly the discussion section.
In abstract part you wrote about antibacterial etc. Should be there is at least one reference in introduction about this part.

Experimental design

• Synergetic experiment it is not clear (in your MS you wrote the value IC50 for menthol and daunorubicin alone but where is the value of menthol + daunorubicin in combination that achieve IC50 for calculation combination index (CI). Did you perform checkerboard assay?
• I observed that you got good result with menthol means there is slightly side effect on normal cells if you can compare this observation with side effect daunorubicin on normal cells it is will improve your study. (If possible do this experiment).

Validity of the findings

no comment

Additional comments

• My queries are
1. What are the possible effects of menthol on host immune cells?
2. For synergistic action of menthol should it be taken as a topical application or oral suspension?
3. Are there other chemotherapy used as treatment of leukemia? If yes, why you are chose daunorubicin?

Annotated reviews are not available for download in order to protect the identity of reviewers who chose to remain anonymous.

Reviewer 3 ·

Basic reporting

The manuscript “Anti-leukemic effect of menthol, a peppermint compound, on induction of apoptosis and autophagy” represents an interesting research work yielding new information about utilisation of menthol in anticancer therapy. There is no doubt that increased number of recent publications concerning the bioactivity of terpenes has highlighted the need for further research on cytostatic, cytotoxic, proapoptotic and autophagy-inducing activity.

Experimental design

Experimental design corresponds to the intention of an experimental in vitro study.

Validity of the findings

Overall, there are no serious concerns or obstacles in manuscript. According to my opinion, the manuscript is logically organized, well written, very well referenced and represents a valuable contribution to the scientific community interested in terpenoid compounds and its determining before consideration can be given to development of the compound for clinical use.

Additional comments

I just suggest several modifications, improvements of the manuscript during follow-up minor revision. In summary, find below my comments, which could help the authors to present their ideas in clearer and more suitable for publication form, during the minor revision of the manuscript.

1) Line 111, cells/ml; line 112 µg/ml and other place in the text of manuscript. “ml” should be “mL”.
2) Line 120 “at 48 h”, line 136 “for 48h”, figure 1, “48 hrs, 48 hrs” Abbreviation for hour should be unified across the manuscript.
3) Line 157, “western blot”, "W" should be as capital letter
4) Figure 1, 2 and Figure 3 legend. And Figure 1, 2 and 2 bar graphs y axis - “Cell viability were measured using MTT assay.” Are the authors convinced that MTT assay in microplate format measure viability only? What about proliferation? Is the designation for “y” axis % cell viability 100% correct?
5) In flow cytometry scatter plots (histograms) presented in figure 4A and same in single parameter histograms of Figure 5A, reader can barley see scaling on x and y axis. Visibility of log number should be improved, or number must be removed from the figure.
6) In figure 4, it seems that treatment considerably increased number necrotic cells. Upper left quadrant in histograms. The percentage share of necrotic cells following exposure should be mentioned in manuscript for MOLT-4 and NB4 cells.

---

## Round 0.2 · accepted · Accept

Thank you for attending to the comments of all three reviewers. I deem these changes to be acceptable and am happy therefore to recommend acceptance.